# CRUSH4SQL: Collective Retrieval Using Schema Hallucination For Text2SQL

**Mayank Kothyari**[*] and **Dhruva Dhingra** and **Sunita Sarawagi**[*] and **Soumen Chakrabarti**
Department of Computer Science and Engineering
Indian Institute of Technology Bombay, Mumbai, India

## Abstract

Existing Text-to-SQL generators require the entire schema to be encoded with the user text. This is expensive or impractical for large databases with tens of thousands of columns. Standard dense retrieval techniques are inadequate for schema subsetting of a large structured database, where the correct semantics of retrieval demands that we rank *sets* of schema elements rather than individual elements. In response, we propose a two-stage process for effective coverage during retrieval. First, we instruct an LLM to hallucinate a minimal DB schema deemed adequate to answer the query. We use the hallucinated schema to retrieve a subset of the actual schema, by composing the results from *multiple* dense retrievals. Remarkably, hallucination — generally considered a nuisance — turns out to be actually useful as a bridging mechanism. Since no existing benchmarks exist for schema subsetting on large databases, we introduce three benchmarks. Two semi-synthetic datasets are derived from the union of schemas in two well-known datasets, SPIDER and BIRD, resulting in 4502 and 798 schema elements respectively. A real-life benchmark called SocialDB is sourced from an actual large data warehouse comprising 17844 schema elements. We show that our method[1] leads to significantly higher recall than SOTA retrieval-based augmentation methods.

## 1 Introduction

State-of-the-art language model based Text-to-SQL generators provide impressive accuracies on well known benchmarks, but they require the entire DB schema as input, along with the user question text (Scholak et al., 2021; Li et al., 2023a; Liu et al., 2023; Rajkumar et al., 2022). Widely-used benchmarks such as SPIDER (Yu et al., 2018) and WikiSQL (Zhong et al., 2017), and even "datasets in the wild," like SEDE (Hazoom et al., 2021), are all associated with modest-sized schema. E.g., the average number of tables and columns in any one schema for SPIDER is 5.3 tables and 28.1 columns, and for SEDE is 29 tables and 212 columns.

In contrast, real-life datasets may have thousands of tables with hundreds of columns per table. E.g., a real-life data warehouse of data about various social indicators of a country comprises of more than 17.8 thousand columns! For such large schema, we cannot afford to include the entire schema in the prompt preceding each query; only a high-recall subset of the schema can be attached to each question. LLM-as-a-service usually charges for each token exchanged between client and server, so we want the subset to be as small as possible while ensuring high recall. Even for in-house (L)LMs or other Text-to-SQL methods, admitting extraneous schema elements as candidates for use in the generated SQL reduces its accuracy (Li et al., 2023a).

Retrieving a subset of a corpus of passages to augment the LLM prompt has become an emerging area (Shi et al., 2023; Ram et al., 2023) for non-Text-to-SQL applications as well. Most retrieval modules depend on standard "dense passage retrieval" (DPR) based on similarity between the question embedding and an embedding of a document (Khattab and Zaharia, 2020) or schema element (Nogueira et al., 2020; Muennighoff, 2022).

We argue (and later demonstrate) that Text-to-SQL needs a more circumspect approach to jointly leverage the strengths of LLMs and dense retrieval. Consider a question *"What is the change in female school enrollment and GDP in Cameroon between 2010 and 2020?"*, to be answered from a large database like World bank data[2]. An effective Text-to-SQL system needs to realize that *GDP* and *female school enrollment* are two key

---

*Corresponding Authors: maykat2017@gmail.com, sunita@iitb.ac.in

[1]The code and dataset for the paper are available at https://github.com/iMayK/CRUSH4SQL

[2]https://datacatalog.worldbank.org/

'atoms' in the question and match these to tables `Development indicators` and `Education statistics` respectively. Additionally, it needs to generalize *Cameroon* to `Country`, and `2010,2020` to `year` to match correct columns in these tables. This requires generalization and phrase-level matching via LLMs (with all the world knowledge they incorporate); pre-deep-NLP and 'hard' segmentation techniques (Gupta and Bendersky, 2015), as well as token-decomposed deep retrieval, such as ColBERT (Khattab and Zaharia, 2020), are unlikely to suffice.

Our setting thus requires us to retrieve, score and select *sets* of schema elements that *collectively* cover or explain the whole query. This bears some superficial similarity with multi-hop question answering (QA). Closer scrutiny reveals that, in multi-hop QA benchmarks such as HotPotQA, each question comes with only 10 passages, out of which 8 are 'distractors' and two need to be selected to extract the answer. The best-performing systems (Li et al., 2023c; Yin et al., 2022) can afford exhaustive pairing of passages along with the question, followed by scoring using all-to-attention — such techniques do not scale to our problem setting with thousands of tables and hundreds of columns per table, where as many as 30 schema elements may be involved in a query.

**Our contributions:** In this paper, we propose a new method called CRUSH[3] that leverages LLM hallucination (generally considered a nuisance) in conjunction with dense retrieval, to identify a small, high-recall subset of schema elements for a downstream Text-to-SQL stage. CRUSH first uses few-shot prompting of an LLM to hallucinate a minimal schema that can be used to answer the given query. In the example above, the LLM might hallucinate a schema that includes tables like
- `Indicators(name, country, year)` and
- `Education enrollment data(type, country, year, value)`

The hallucinated schema contains strings that are significantly closer to the gold table names mentioned earlier. We use the hallucinated schema elements to define a collection of index probes for fast index-based retrieval of the actual schema elements in the DB. Finally, CRUSH approximately solves a novel combinatorial subset selection objective to determine a high-recall, small-sized schema subset. The objective includes special terms to

---

[3]Collective Retrieval Using Schema Hallucination

maximize coverage of distinct elements of the hallucinated schema while rewarding connectivity of the selected subset in the schema graph.

Our second contribution involves the creation of three novel benchmarks for the task of retrieval augmentation related to Text-to-SQL conversion on a large schema. We developed two semi-synthetic benchmarks, encompassing 4502 and 768 columns respectively, by taking a union of all databases from the well-known SPIDER benchmark, as well as the relatively recent BIRD benchmark. The third benchmark is sourced from a production data warehouse and features *17.8 thousand columns*. This serves to mitigate a critical limitation in existing Text-to-SQL benchmarks, which have much smaller schema. Beyond its large scale, our third benchmark introduces additional challenges such as a significantly higher overlap in column names, and challenging lexical gaps (when compared to SPIDER), between schema mentions in the question and the actual schema name.

Using these benchmarks, we present an extensive empirical comparison between CRUSH and existing methods. We show consistent gains in recall of gold schema elements, which translates to increased accuracy of Text-to-SQL generation. The results of our analysis provide valuable insights into the weaknesses of the existing single-embedding or token-level representations.

## 2 Notation and problem statement

We are given a large database schema $\mathcal{D}$ consisting of a set of tables $\mathcal{T}$, with each table $t \in \mathcal{T}$ comprising of a set of columns $c \in \mathcal{C}(t)$. We will use $d$ to denote a schema element ('document'), which can be either a table or a column. A schema element $d$ has a textual name or description $\mathcal{S}(d)$. The text associated with a column $t.c$ is written as the concatenation $\mathcal{S}(t).\mathcal{S}(c)$ as shown in Figure 1.

Apart from the database, the input includes a natural language question $x$. Question $x$ is associated with a (possibly unknown) correct ('gold') SQL query $q(x)$. The gold SQL query $q(x)$ mentions a subset $R(q(x))$ of the schema elements from $\mathcal{D}$. Almost always, $|R(q(x))| \ll |\mathcal{D}|$.

Our goal is to retrieve from $\mathcal{D}$, a (small) subset $R(x) \subset \mathcal{D}$ that includes $R(q(x))$, i.e., $R(q(x)) \subseteq R(x)$. The question $x$ will be concatenated with the schema subset $R(x)$ and input to a Text-to-SQL model to convert the question into an SQL query. There are multiple reasons to minimize $|R(x)|$.

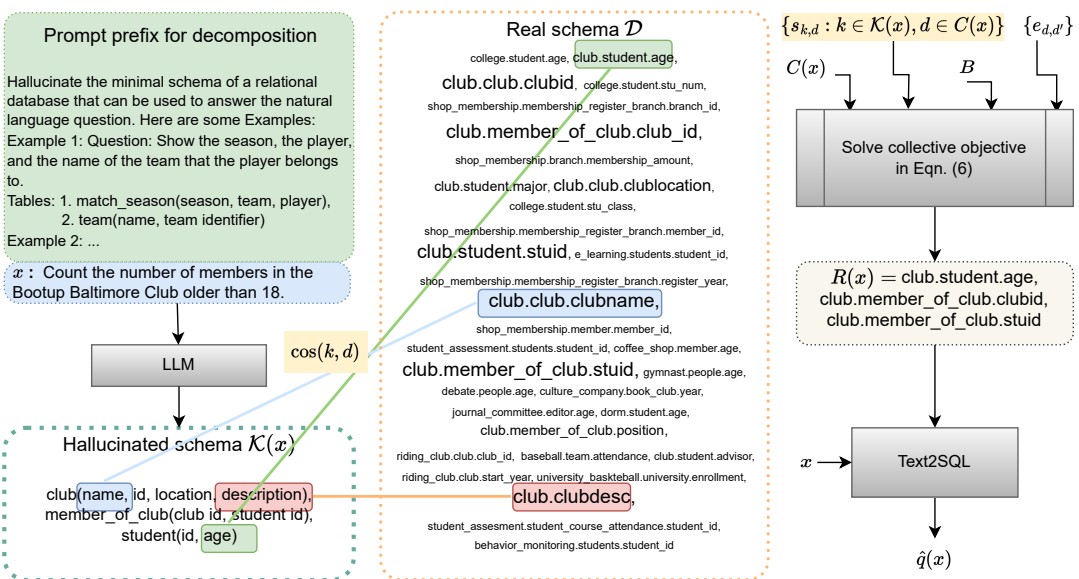

Figure 1: Illustration of how CRUSH works. We prefix a decomposition prompt with in-context examples to the question $x$ and submit to an LLM. The response has a hallucinated schema $\mathcal{K}(x)$ with 'query elements' $k$. The real schema $\mathcal{D}$ has 'documents' $d$, which may be connected to each other via same-table-as and primary-foreign-key relations $e_{d,d'}$. The queries from the hallucinated schema are used to get candidate real schema elements $C(x)$. Similarities $\cos(\boldsymbol{k}, \boldsymbol{d})$ lead to $s_{k,d}$ scores. $B$ is the budget size of $R(x)$, the relevant real schema subset to be returned by the optimizer. $R(x)$ is used by the downstream Text-to-SQL system.

(1) The size of $\mathcal{D}$ is significantly larger than can be fitted in a prompt to a LLM. (2) LLM-as-a-service usually charges usage fees in proportion to the number of tokens exchanged with the client. (3) The performance of even in-house (L)LMs or other Text-to-SQL systems degrade when extraneous schema elements are presented as possible candidates for inclusion in the SQL query. In this paper we focus on the task of efficiently retrieving $R(x)$ from $\mathcal{D}$, while maximizing the recall of gold schema elements in $R(q(x))$.

We assume that the schema $\mathcal{D}$ is indexed in a pre-processing step. Each table $t(c_1, \ldots)$ in the database $\mathcal{D}$ is exploded into the form $t.c$ for each column, where '.' is a separator character. Each column now acts as a 'document' $d$ in an information retrieval system, except that we will retrieve *sets* of documents. Each document is sent into a pre-trained transformer, SGPT (Muennighoff, 2022), or the LLM service to get an embedding $\boldsymbol{d}$.

## 3 The CRUSH4SQL Approach

Successfully matching a user question $x$ to a relevant schema $R(q(x))$ could be quite non-trivial, since the schema element names or descriptions are often not directly mentioned in the question. For example, consider the question $x$ = *Count the number of members in the Bootup Baltimore Club older than 18.* In order to match this question prop-

erly to gold schema $\mathcal{D}$, containing such schema elements as Age, Club description, and Club members, etc., we need to perform multiple types of lexical and syntactic reasoning on the question and schema:

- extract segments like *older than 18* and *Bootup Baltimore Club*,
- generalize the first segment to a likely schema element Age, and,
- for the second segment, instead of attempting to match strings like *Bootup Baltimore* verbatim, match to a schema element called Club description.

To bridge such large lexical gap between the tokens of $x$ and the gold schema $R(q(x))$, we designed a two-phase approach. In the first phase, we transform $x$ into an intermediate form $\mathcal{K}(x)$ comprising of multiple generalized segments leveraging an LLM. In the second phase, we use $\mathcal{K}(x)$ to retrieve from the DB schema $\mathcal{D}$ suitable schema element subset $R(x)$, by approximately optimizing a combinatorial objective that collectively maximizes coverage of all elements of $\mathcal{K}(x)$. We describe these two phases next.

### 3.1 LLM-based Query Transformation

Our goal here is to infer from the question text $x$, a set $\mathcal{K}(x)$ of intermediate search strings, which, when used to probe a suitable index over the client's

DB schema, will retrieve $R(x)$. After unsuccessful attempts with query decomposition and token-level retrieval methods, we proposed to harness an LLM to aid with this task.

Initially, we attempted to use the LLM to extract variable name mentions or perform conventional query decomposition like in (Pereira et al., 2022). While these showed promise, we obtained much better results when we took a leap and harnessed the power of an LLM to directly hallucinate a DB schema that could be used to answer the question $x$.

We use state-of-the-art LLMs with few-shot prompting to hallucinate such a schema. We employ GPT-3 (text-davinci-003) with a fixed prompt comprising of six in-context examples as shown in the first half of Table 1 for one of our datasets. We create the desired output $\mathcal{K}(x)$ corresponding to each $x$ in the prompt guided by the gold schema $R(q(x))$ in the SQL corresponding to $x$. In the second half of Table 1 we show examples of a few hallucinated schema from the LLM in response to the prompt. In each of the four examples, we see that the LLM has produced very reasonable schemas applying the right level of generalization to constants and implicitly segmenting the query across multiple tables in the hallucinated schema. The same trend holds for the significantly more challenging examples from the SocialDB dataset as seen in Table 15, and is also observed in the BirdUnion dataset as shown in Table 16.

We experimented with a few other prompt types, and we will present a comparison in Section 5.4 of our empirical evaluation.

## 3.2 Collective Retrieval

In this stage our goal is to retrieve a subset $R(x)$ from $\mathcal{D}$ so that collectively $R(x)$ is closest to the halluncinated schema $\mathcal{K}$ in the context of $x$. First, we retrieve a candidate set $C(x)$ using $\mathcal{K}$ as probes on the indexed embeddings in $\mathcal{D}$, and then we collectively match $C(x)$ to $\mathcal{K}(x)$.

### 3.2.1 Retrieving candidate set $C(x)$

$\mathcal{K}(x)$ consists of a set of hallucinated tables with their hallucinated columns, each of the form $t(c_1, \ldots)$. These are written out as a set of "$t.c_1$" column names prefixed with table names. Henceforth, we regard $\mathcal{K}(x)$ as a set of such hallucinated texts $\{k\}$. Each $k \in \mathcal{K}(x)$ is converted into an embedding vector $\boldsymbol{k}$ for retrieving 'real' schema elements, via the following steps:

1: form concatenation "$x\ t.c$"

2: apply a pretrained transformer, SGPT (Muennighoff, 2022)

3: get per-token contextual embedding vectors

4: average-pool per-token embeddings into $\boldsymbol{k}$

(Through experiments, we will show that using the form "$x\ t.c$" improves recall, compared to not prefixing $x$.) At this stage, $\mathcal{K}(x)$ has been converted into a bag of vectors $\{\boldsymbol{k}\}$. We perform a nearest neighbor search on $\mathcal{D}$ using each key vector $\boldsymbol{k}$, and retain some number of top matches per probe $\boldsymbol{k}$ based on cosine similarity of their embeddings. This gives us the candidate set $C(x)$ of schema elements from $\mathcal{D}$. See Figure 1.

**Cosine baseline:** A baseline method may simply return $C(x)$. However, $C(x)$ thus collected was observed to be biased toward generic, uninformative schema names such as Name and Identifier that appear across many tables. This hurts coverage. Therefore, we design a more careful optimization around $C(x)$, described next.

### 3.2.2 Retrieval objective

We extract from the candidate set $C(x)$ a manageable subset $R(x)$ with size $|R(x)| \leq B$ for some size budget $B$, that provides coverage to all parts of $\mathcal{K}$, and also to reward connectivity in the schema graph on the retrieved subset. Recall that a large budget not only results in a higher expense to use an LLM-as-a-service, but may also provide a more challenging generation task for the downstream Text-to-SQL module.

**Entropy-guided similarity:** Instead of just cosine as the distance between a $k \in \mathcal{K}$ and $d \in \mathcal{D}$, we refine the similarity to score match of rarer columns higher. Consider some $k \in \mathcal{K}(x)$ that has good matches with many schema elements $d \in C(x)$. In information retrieval, this is analogous to a query word with low inverse document frequency (IDF) (Manning et al., 2008), and its effect on scores should be damped down. In the deep retrieval regime, we use a notion of *entropy* to act as a surrogate for IDF. Specifically, let $\cos(\boldsymbol{k}, \boldsymbol{d})$ be the cosine score between $k$ and $d$. Fix $k$ and consider the multinomial probability distribution

$$\left\{ \frac{\frac{1}{2}(1 + \cos(\boldsymbol{k}, \boldsymbol{d}))}{\sum_{d' \in C(x)} \frac{1}{2}(1 + \cos(\boldsymbol{k}, \boldsymbol{d}'))} : d \in C(x) \right\} \quad (1)$$

in the multinomial simplex $\Delta^{|C(x)-1|}$. This multinomial distribution has an entropy, which we will denote by $H(k)$. If $H(k)$ is large, that means $k$ has no sharp preference for any schema element

in $C(x)$, so its impact on the perceived similarity $\cos(\boldsymbol{k}, \boldsymbol{d})$ should be dialed down, inspired by TFIDF vector space model from information retrieval. We achieve this effect via the score

$$s_{k,d} = \frac{1}{2}(1 + \cos(\boldsymbol{k}, \boldsymbol{d}))\, \sigma(\bar{H} - H(k)), \quad (2)$$

where $\sigma(\cdot)$ is the standard sigmoid shifted by the average entropy $\bar{H}$ defined as the average $H(k)$ over all $k \in \mathcal{K}$.

**Coverage score:** We will assume that the hallucinated schema elements are all informative, so we would like to 'cover' all of them using $R(x)$. We score coverage of a $k$ by $R(x)$ using a soft maximization function defined as

$$\mathrm{smx}(s_{kd} : d \in R(x)) = \log \sum_{s_{kd}} \exp(s_{kd}) \quad (3)$$

The first part of our collective objective is

$$O_1(R(x)) = \sum_{k \in \mathcal{K}(x)} \mathrm{smx}(s_{kd} : d \in R(x)) \quad (4)$$

**Connections between 'documents':** Schema elements are not isolated, but related to each other using the rich structure of the schema graph (Wang et al., 2020, Figure 2). For example, two columns may belong to the same relation, or they may belong to different tables connected by a foreign key-primary key link. A scalar connectivity score $e(d, d') \geq 0$ characterizes the strength of such a connection between $d, d' \in C(x)$. In our experiments we choose $e(d, d')$ to be a non-zero constant $\gamma$ for all column pairs that are part of the same table or connected by a foreign key. We should choose $R(x)$ to also maximize

$$O_2(R(x)) = \sum_{d \in R(x)} \mathrm{smx}(e(d, d') : d' \in R(x)) \quad (5)$$

The function smx is chosen instead of directly summing $e(d, d')$ for all pairs in $R(x)$ to prevent quadratically growing rewards for large subgraphs.

**Overall objective:** Combining the two desiderata, we get our overall objective as

$$R(x) = \operatorname*{argmax}_{\substack{R \subseteq C(x) \\ |R|=B}} \Big( O_1(R) + \clubsuit\, O_2(R) \Big), \quad (6)$$

with a balancing hyperparameter ♣. It is possible to express the above optimization as a mixed integer linear program. In practice, we find it expeditious to use a simple greedy heuristic. Also, we fix ♣ = 1 in all experiments.

## 4  Related work

**Dense Passage Retrieval (DPR):** A default method of identifying $R(x)$ is based on nearest neighbors in a dense embedding space. First a language model (LM) $M$ like SGPT (Muennighoff, 2022) or OpenAI's Similarity and Search Embeddings API (model text-embedding-ada-002) converts each string $\mathcal{S}(d)$ into a fixed-length embedding vector which we denote $\boldsymbol{d} = M(\mathcal{S}(d))$. Given a question $x$, we obtain the embedding $\boldsymbol{x}$ of $x$, i.e., $\boldsymbol{x} = M(x)$, and then retrieve the $K$ nearest neighbors of $\boldsymbol{x}$ in $\mathcal{D}$ as $R(x) = \text{K-NN}(\boldsymbol{x}, \{\boldsymbol{d} : d \in \mathcal{D}\})$. This method has the limitation that the top-K retrieved elements may be skewed towards capturing similarity with only a subset of the gold schema $R(q(x))$. The RESDSQL (Li et al., 2023a) Text-to-SQL system, too, selects the most relevant schema items for the encoder, using a cross-encoder (based on RoBERTa) between the question and the whole schema. Although the early-interaction encoder may implicitly align question tokens to schema tokens, there is no apparent mechanism to ensure phrase discovery and (LLM knowledge mediated) measurement of coverage of question phrase with schema element descriptions.

**LLMs for ranking:** Sun et al. (2023) use an LLM for relevance ranking in information retrieval. They show that a properly instructed LLM can beat supervised learning-to-rank approaches. No query decomposition of multi-hop reasoning is involved. Other uses of LLMs in scoring and ranking items are in recommender systems (Gao et al., 2023; Hou et al., 2023): turning user profiles and interaction history into (recency focused) prompts, encoding candidates and their attributes into LLM inputs, and asking the LLM to rank the candidates. Notably, LLMs struggle to rank items as the size of the candidate set grows from 5 to 50.

**LLMs for question decompostion and retrieval:** Decomposing complex questions has been of interest for some years now (Wolfson et al., 2020). In the context of LLMs, Pereira et al. (2022) propose a question answering system where supporting evidence to answer a question can be spread over multiple documents. Similar to us, they use an LLM to decompose the question using five in-context examples. The subquestions are used to retrieve passages. A more general architecture is proposed by Khattab et al. (2022). None of these methods use the notion of a hallucinated schema. They match subquestions to passages, not structured schema elements. They entrust the final collective answer extraction to an opaque LM. In contrast, we have an interpretable collective schema selection step.

| # | LLM prompt: Hallucinate a minimal schema of a relational database that can be used to answer the natural language question. Here are some examples: |
|---|---|
| $x$ | Count the number of members in the Bootup Baltimore Club older than 18. |
| $\mathcal{K}$ | Club(Name, id, description, location), member_of_club(club id, student id), Student(id, age) |
| $x$ | What are the names of all stations with a latitude smaller than 37.5? |
| $\mathcal{K}$ | Station(Name, Latitude) |
| $x$ | Show the season, the player, and the name of the team that players belong to. |
| $\mathcal{K}$ | Match_season(season, team, player), Team(name, team identifier) |
| $x$ | Find the first name and age of the students who are playing both Football and Lacrosse. |
| $\mathcal{K}$ | SportsInfo(sportname, student id), Student(age, first name, student id) |
| $x$ | What are the names of tourist attractions reachable by bus or is at address 254 Ottilie Junction? |
| $\mathcal{K}$ | Locations(address, location id), Tourist_attractions(how to get there, location id, name) |
| $x$ | Give the name of the highest paid instructor. |
| $\mathcal{K}$ | Instructor(Name, Salary) |
| **Hallucinated $\mathcal{K}$ generated by LLM given input $x$** | |
| $x$ | What are the names of properties that are either houses or apartments with more than 1 room? |
| $\mathcal{K}$ | Property(name, type, number of rooms) |
| $x$ | Which employee received the most awards in evaluations? Give me the employee name. |
| $\mathcal{K}$ | Employee(name, employee id), Evaluations(employee id, awards) |
| $x$ | What is the document name and template id with description with the letter 'w' in it? |
| $\mathcal{K}$ | Document(name, description, template id) |
| $x$ | What semester ids had both Masters and Bachelors students enrolled? |
| $\mathcal{K}$ | Semester(id, start date, end date), Enrollment(semester id, student id, degree), Student(id, name) |

Table 1: Examples of in-context training examples given to the LLM to prompt it to hallucinate a minimal schema of a database that can be used to answer the given question.

# 5 Experiments

In this section we compare CRUSH with existing methods for schema subsetting. We also present a detailed ablation on the various design options for CRUSH.

## 5.1 Datasets

We test on the following two benchmarks that we designed, because of the absence of any pre-existing large-schema benchmark.

**SpiderUnion:** This is a semi-synthetic benchmark derived from SPIDER (Yu et al., 2018), a popular Text-to-SQL benchmark, where each question $x$ corresponds to one of the 166 database schemas. These schemas cover various, somewhat overlapping topics such as Singers, Concert, Orchestra, Dog Kennels, Pets, and so forth. Each schema is compact, averaging approximately 5.3 tables and 28 columns. To simulate a larger schema, we combine these 166 schemas, prefixing each table name with its corresponding database name. The unified schema comprises 4,502 columns distributed across 876 tables, forming the $\mathcal{D}$. We evaluate the model using 658 questions sourced from the SPIDER development set after excluding questions where the gold SQL contains a '$\star$'. Unlike in SPIDER, the question $x$ is not associated with one of these 166 DB ids. Instead, the system has to find the correct schema subset.

Our evaluation metric, given a retrieved set $R(x)$, is *recall*, defined as $\frac{|R(q(x)) \cap R(x)|}{|R(q(x))|}$. For each question in the test set, since the gold SQL is available, we have access to the gold retrieval set $R(q(x)) \subset \mathcal{D}$. We measure recall only over column names, since if a column name is selected in $R(x)$, the table name is always implicitly selected.

**BirdUnion:** Following the same approach as with SpiderUnion, we created BirdUnion from BIRD (Li et al., 2023b), a relatively new cross-domain dataset, with 95 large databases and a total size of 33.4 GB. It covers more than 37 professional domains, such as blockchain, hockey, healthcare, education, etc. To simulate a larger schema, we combined 11 schemas in the development set (where each schema covers approximately 6.82 tables and 10.64 columns), prefixing each table name with its corresponding database name. The unified schema comprises 798 columns distributed across 75 tables, constituting $\mathcal{D}$. We evaluate the model using 1534 questions sourced from BIRD dev-set. We adopt the same evaluation metric as with SpiderUnion.

**SocialDB:** We created this benchmark from a real-life data warehouse, which collates statistics on various social, economic, and health indicators from a large country, featuring diverse geographical and temporal granularities. The complete database schema is publicly accessible, though we withhold the URL during the anonymity period. The warehouse holds approximately 1046 tables and a total of 18,685 columns. Each table and column carries descriptive names. From administrators of the Web-

site, we obtained 77 questions along with the gold tables, which contain the answers. Some examples of questions and schema names can be found in Table 15. We could not obtain gold column names on this dataset, and thus our evaluation metric is table recall measured as $\frac{|\text{Tables}(q(x)) \cap \text{Tables}(R(x))|}{|\text{Tables}(q(x))|}$.

## 5.2 Methods Compared

**Single DPR (SGPT):** This is the popular Dense Passage Retrieval (DPR) baseline where we use the SGPT LM (Muennighoff, 2022) to embed $x$ into a single embedding vector $\boldsymbol{x}$ and retrieve from $\mathcal{D}$ based on cosine similarity with a $\boldsymbol{d} \in \mathcal{D}$.

**Single DPR (OpenAI):** As above, except we use OpenAI's Similarity and Search Embeddings API (text-embedding-ada-002) as the LM.

**Token-level Embedding (ColBERT):** Instead of searching with a single embedding we perform token-decomposed retrieval for finer-grained interaction (Khattab and Zaharia, 2020).

**CRUSH:** Our proposed method, where we use OpenAI's DaVinci to get hallucinated schema with prompts in Table 1 for SpiderUnion and Table 15 for SocialDB. Embeddings are obtained using SGPT, because we need contextual embeddings for tokens in $x$ (not supported in OpenAI's embedding API) to build embeddings for $k \in \mathcal{K}$. The default candidate set size is limited to 100, with edge scores assigned a default value of $e(d, d') = 0.01$ for all edges in both SpiderUnion and BirdUnion, while it is zero for SocialDB.

## 5.3 Overall Comparison

In Table 2 we present recall for different budgets on the size of the retrieved set $R(x)$ on the four different methods on both the SpiderUnion and SocialDB datasets. Key observations:

- We see a significant boost in the recall by CRUSH, particularly at low to medium budget levels. For example, on SpiderUnion at a budget of ten columns, we recall 83% of gold columns whereas the best existing method only gets up to 77%. On the BirdUnion dataset, we recall 76% with CRUSH, while the best alternative method reaches only 56%. On the more challenging SocialDB dataset we recall 58% of gold tables whereas the best alternative method gets 49%.
- Token-level methods are worse than even Single DPR-based methods.
- Embedding of OpenAI is slightly better than than of SGPT.

In Table 14 we present anecdote that illustrates how CRUSH is able to more effectively cover the gold schema, compared to Single DPR. Single DPR gets swamped by matches to the student_id column across many diverse tables, whereas CRUSH is able to cover all gold columns.

**Impact of improved recall on Text-to-SQL generation accuracy:** We use the state-of-art RES-DSQL (Li et al., 2023a) model for generating the SQL using the schema subset selected by various systems. Following standard practice, we use Exact Match (EM) and Execution Match (EX) accuracy to evaluate the quality of the generated SQL. As seen in Table 3, the improved recall of schema subsetting translates to improved accuracy of Text-to-SQL generation. However, beyond a budget of 30, we see a *drop* in the accuracy of the generated SQL, presumably because the Text-to-SQL method gets distracted by the irrelevant schema in the input. For the BirdUnion dataset, the RESDSQL system could not handle the larger schema at budget 100, but we expect a similar trend.

## 5.4 Robustness Across Prompt Variations

Before we arrived at the schema hallucination approach, we experimented with other prompts motivated by the techniques in the question decomposition literature. In Table 4 we present two such alternatives. (1) Variables: that seeks to identify key isolated variables mentioned in $x$, and (2) Relations: that relates the variables to a subject, roughly analogous to a table name. We present a comparison of different prompt types in Table 6. Observe that the schema-based prompts of CRUSH are consistently better than both the earlier prompts.

In CRUSH we used six examples of hallucinated schema in the prompt for in-context learning. We reduce that number to two and four and present the results in Table 7. Recall improves significantly with more examples but even with two examples, the LLM provides gains over the baseline.

In Tables 8 and 9, we present the impact of random shuffling on the six in-context examples in the prompt (Tables 1 and 15). We provide mean and standard deviation schema retrieval recall and downstream Text-to-SQL generation accuracy. Notably, across the five iterations, the standard deviation in recall and EX remains low, suggesting robust performance regardless of the prompt order. In Tables 10 and 11, we investigate the robustness of recall and EX values when different sets of few-

| Data set | Method | Budget$\longrightarrow$ | | | | | | |
|---|---|---|---|---|---|---|---|---|
| | | r @ 3 | r @ 5 | r @ 10 | r @ 20 | r @ 30 | r @ 50 | r @ 100 |
| SpiderUnion | Single DPR (SGPT) | 0.50 | 0.61 | 0.76 | 0.86 | 0.89 | 0.92 | 0.95 |
| | Single DPR (OpenAI) | 0.55 | 0.64 | 0.77 | 0.86 | 0.90 | 0.93 | 0.96 |
| | Token-level (ColBERT) | 0.49 | 0.59 | 0.72 | 0.84 | 0.88 | 0.92 | 0.95 |
| | CRUSH (ours) | **0.59** | **0.72** | **0.83** | **0.90** | **0.92** | **0.94** | **0.97** |
| SocialDB | Single DPR (SGPT) | 0.35 | 0.40 | 0.45 | 0.52 | 0.58 | 0.67 | 0.73 |
| | Single DPR (OpenAI) | 0.39 | 0.44 | 0.49 | 0.56 | 0.60 | 0.67 | **0.76** |
| | Token-level (ColBERT) | 0.36 | 0.36 | 0.44 | 0.51 | 0.55 | 0.57 | 0.64 |
| | CRUSH (ours) | **0.40** | **0.52** | **0.58** | **0.67** | **0.69** | **0.71** | 0.75 |
| BirdUnion | Single DPR (OpenAI) | 0.33 | 0.43 | 0.56 | 0.66 | 0.72 | 0.8 | 0.93 |
| | CRUSH (ours) | **0.39** | **0.54** | **0.71** | **0.82** | **0.88** | **0.92** | **0.97** |

Table 2: Comparison of recall for various column ranking methods on the test sets of SpiderUnion, SocialDB, and BirdUnion datasets. We observe a significant boost in the recall by CRUSH of the gold schema at all budget levels.

| Data set | Method | Budget$\longrightarrow$ | | | | | | |
|---|---|---|---|---|---|---|---|---|
| | | r @ 3 | r @ 5 | r @ 10 | r @ 20 | r @ 30 | r @ 50 | r @ 100 |
| SpiderUnion | Single DPR (OpenAI) | 0.29/0.33 | 0.35/0.41 | 0.45/0.53 | 0.48/0.57 | 0.48/0.59 | 0.48/0.59 | 0.51/0.62 |
| | CRUSH (ours) | **0.35/0.39** | **0.46/0.53** | **0.52/0.60** | **0.53/0.64** | **0.54/0.64** | **0.52/0.63** | **0.52/0.62** |
| BirdUnion | Single DPR(OpenAI) | 0.03/0.07 | 0.05/0.10 | 0.07/0.13 | 0.09/0.15 | 0.09/0.16 | 0.10/0.18 | -/- |
| | CRUSH (ours) | **0.04/0.07** | **0.07/0.11** | **0.09/0.15** | **0.11/0.19** | **0.11/0.19** | **0.12/0.21** | -/- |

Table 3: Exact Match (EM) / Execution Match (EX) accuracy when RESDSQL is used to generate SQL on schema retrieved at various budgets from CRUSH and Single DPR (OpenAI). The higher recall of CRUSH's retrievals lead to more accurate SQLs. Very large budget worsens SQL accuracy.

| Example | Question |
|---|---|
| $x$ | Count the number of members in club Bootup Baltimore older than 18. 
 Variables: Age, Club 
 Relations: Age of club members, Name of the club |
| $x$ | What are the names of all stations with a latitude smaller than 37.5? 
 Variables: Latitude, Station 
 Relations: Latitude of the station, Name of the station |
| $x$ | Show the season, the player, and the name of the team that players belong to. 
 Variables: Match, Season, Player 
 Relations: Name of the team the player belongs to, Season(s) played by player, Name of the player |
| $x$ | Find the first name and age of the students who are playing both Football and Lacrosse. 
 Variables: Student, Age, Game, Football 
 Relations: Sports played by student, Age of student, Name of student |

Table 4: Two alternative forms of transforming $x$ into segments. Contrast these with the hallucinated schema in Table 1. Only top-4 shown due to lack of space.

shot in-context examples. Low standard deviation suggests a high level of robustness across different samples. In Table 12, we explore the LLM's schema generation under different temperature settings, finding minimal impact on the results. Finally, as shown in Table 13, even in the zero-shot setting, CRUSH exhibits significant improvement over the baseline Single DPR based retrieval.

## 5.5 Ablation on Collective Retriever

CRUSH includes a number of careful design choices. In Table 5 we show the impact of each design choice.

- During retrieval, we obtain the embeddings of a $k \in \mathcal{K}$ jointly with $x$. In contrast, if we independently embed $k$, the recall drops significantly.
- After retrieval, the overall objective of collective selection (Eq 6) incorporates three key ideas:

entropy guided similarity, edge scores, and coverage of hallucinated schema elements. We study the impact of each. We remove the entropy discounting in Eqn. (2), and observe a drop in recall at low budget levels. When we remove the edge scores, we also see a mild drop.

- To study the impact of coverage, we replace the soft-max function $\mathrm{smx}()$ with a simple summation so that for each selected $d \in \mathcal{D}$, the reward is just the sum of similarity to each $k \in \mathcal{K}$. We find that the recall suffers. A coverage encouraging objective is important to make sure that the selected items are not over-represented by matches to a few $k \in \mathcal{K}$.

## 6 Conclusion

While LLMs incorporate vast world knowledge and corpus statistics, they may be unfamiliar with

|  | Budget | | | | | | |
|---|---|---|---|---|---|---|---|
|  | r @ 3 | r @ 5 | r @ 10 | r @ 20 | r @ 30 | r @ 50 | r @ 100 |
| CRUSH | 0.59 | 0.72 | 0.83 | 0.90 | 0.92 | 0.94 | 0.97 |
| $-$ $x$-contextual embedding | 0.53 | 0.66 | 0.77 | 0.86 | 0.90 | 0.93 | 0.95 |
| $-$ Entropy | 0.54 | 0.67 | 0.81 | 0.89 | 0.91 | 0.94 | 0.97 |
| $-$ Edge scores | 0.57 | 0.71 | 0.83 | 0.90 | 0.92 | 0.95 | 0.97 |
| $-$ Coverage | 0.54 | 0.67 | 0.81 | 0.89 | 0.91 | 0.94 | 0.97 |

Table 5: Ablation on design choices of CRUSH on the SpiderUnion dataset. Each row after the first, provides CRUSH with one of the key design elements of CRUSH removed.

| Prompt type | Budget | | | | |
|---|---|---|---|---|---|
|  | r @ 3 | r @ 5 | r @ 10 | r @ 20 | r @ 30 |
| Variables | 0.43 | 0.56 | 0.73 | 0.84 | 0.89 |
| Relations | 0.57 | 0.66 | 0.80 | 0.89 | 0.92 |
| CRUSH | 0.57 | 0.71 | 0.83 | 0.90 | 0.92 |

Table 6: Effect of prompt types. Compared to the CRUSH prompt that ask for hallucinating a schema, the earlier two prompts, motivated by traditional question decomposition viewpoint, are much worse.

| Number of prompts | Budget | | | | |
|---|---|---|---|---|---|
|  | r @ 3 | r @ 5 | r @ 10 | r @ 20 | r @ 30 |
| Single DPR (SGPT) | 0.50 | 0.61 | 0.76 | 0.86 | 0.89 |
| CRUSH (2 shots) | 0.52 | 0.67 | 0.81 | 0.88 | 0.90 |
| CRUSH (4 shots) | 0.58 | 0.70 | 0.82 | 0.89 | 0.92 |
| CRUSH (6 shots) | 0.59 | 0.72 | 0.83 | 0.90 | 0.92 |

Table 7: Number of shots (in-context examples) in CRUSH prompts. Even two examples give gains over the baseline and increasing the number of examples improves recall.

| Dataset | Budget | | | | |
|---|---|---|---|---|---|
|  | r @ 3 | r @ 5 | r @ 10 | r @ 20 | r @ 30 |
| SpiderUnion | | | | | |
| Recall Mean | 0.58 | 0.70 | 0.82 | 0.89 | 0.92 |
| Recall Std | 0.01 | 0.01 | 0.00 | 0.00 | 0.01 |
| SocialDB | | | | | |
| Recall Mean | 0.37 | 0.47 | 0.56 | 0.65 | 0.69 |
| Recall Std | 0.03 | 0.03 | 0.04 | 0.02 | 0.02 |

Table 8: Effect of randomly shuffling (in-context) examples in CRUSH prompts (SpiderUnion) over recall.

(possibly private) client DB schemas, which can be very large, rendering impractical or expensive any attempt to upload the full schema in-context along with questions for Text-to-SQL applications. Remarkably, we find a workable middle ground by allowing the LLM to hallucinate a schema from the question and limited in-context examples with no reference to the client schema. Then we formulate a novel collective optimization to map the hallucinated schema to real DB schema elements. The resulting real schema subset that is retrieved has a small size, yet high recall This schema subset can be readily uploaded to (L)LM-based Text-to-SQL methods. The reduced space of client DB schema elements also improves the accuracy of generated SQL for state-of-the-art Text-to-SQL implementa-

tions.

| Dataset | Budget | | | | |
|---|---|---|---|---|---|
|  | r @ 3 | r @ 5 | r @ 10 | r @ 20 | r @ 30 |
| EM Mean | 0.33 | 0.45 | 0.51 | 0.51 | 0.51 |
| EM Std | 0.02 | 0.00 | 0.01 | 0.01 | 0.02 |
| EX Mean | 0.38 | 0.52 | 0.60 | 0.61 | 0.60 |
| EX Std | 0.01 | 0.00 | 0.01 | 0.02 | 0.02 |

Table 9: Effect of randomly shuffling (in-context) examples in CRUSH prompts (SpiderUnion) over EM/EX.

| Dataset | Budget | | | | |
|---|---|---|---|---|---|
|  | r @ 3 | r @ 5 | r @ 10 | r @ 20 | r @ 30 |
| Recall Mean | 0.60 | 0.70 | 0.80 | 0.90 | 0.90 |
| Recall Std | 0.02 | 0.01 | 0.01 | 0.01 | 0.01 |

Table 10: Effect of selecting different (in-context) examples in CRUSH prompts (SpiderUnion) over recall.

| Dataset | Budget | | | | |
|---|---|---|---|---|---|
|  | r @ 3 | r @ 5 | r @ 10 | r @ 20 | r @ 30 |
| EM Mean | 0.34 | 0.46 | 0.52 | 0.52 | 0.51 |
| EM Stdev | 0.01 | 0.02 | 0.01 | 0.02 | 0.02 |
| EX Mean | 0.38 | 0.53 | 0.60 | 0.62 | 0.61 |
| EX Stdev | 0.01 | 0.01 | 0.01 | 0.02 | 0.02 |

Table 11: Effect of selecting different (in-context) examples in CRUSH prompts (SpiderUnion) over EM/EX.

| Dataset | Budget | | | | |
|---|---|---|---|---|---|
|  | r @ 3 | r @ 5 | r @ 10 | r @ 20 | r @ 30 |
| SpiderUnion | | | | | |
| Recall at temp = 0 | 0.59 | 0.72 | 0.83 | 0.90 | 0.92 |
| Recall at temp = 0.5 | 0.58 | 0.70 | 0.82 | 0.89 | 0.92 |
| Recall at temp = 1 | 0.58 | 0.69 | 0.82 | 0.89 | 0.91 |
| SocialDB | | | | | |
| Recall at temp = 0 | 0.40 | 0.52 | 0.58 | 0.69 | 0.71 |
| Recall at temp = 0.5 | 0.41 | 0.50 | 0.61 | 0.67 | 0.71 |
| Recall at temp = 1 | 0.36 | 0.47 | 0.56 | 0.63 | 0.70 |

Table 12: Effect of temperature changes on recall.

| Dataset | Budget | | | | |
|---|---|---|---|---|---|
|  | r @ 3 | r @ 5 | r @ 10 | r @ 20 | r @ 30 |
| SpiderUnion | | | | | |
| Zero shot | 0.58 | 0.72 | 0.84 | 0.90 | 0.92 |
| Single DPR(OpenAI) | 0.55 | 0.64 | 0.77 | 0.86 | 0.90 |
| CRUSH (ours) | 0.59 | 0.72 | 0.83 | 0.90 | 0.92 |
| SocialDB | | | | | |
| zero shot | 0.33 | 0.43 | 0.52 | 0.65 | 0.68 |
| Single DPR(OpenAI) | 0.39 | 0.44 | 0.49 | 0.56 | 0.60 |
| CRUSH (ours) | 0.40 | 0.52 | 0.58 | 0.67 | 0.69 |

Table 13: Effect of zero-shot prompting on recall.

# 7 Limitations

Removing two limitations in CRUSH may be useful in future work. First, at present, hallucination is almost completely unguided by the client DB schema. It would be of interest to explore if the client DB schema can be compressed into a very small prompt text to give some limited guidance to the LLM schema hallucinator. At present the link weights $e(d, d')$ between schema elements $d, d'$ are hardwired; it may be useful to extend the learning optimization to fit these weights.

# 8 Acknowledgement

We thank Microsoft for sponsoring access to Azure OpenAI API via the Accelerate Foundation Models Academic Research Initiative. A special thanks to Niti Aayog's NDAP team for letting us use their data schema and sharing their query workload. Additionally, we extend our thanks to Mayur Datar and Vinayak Borkar for stimulating discussions. We thank IBM's AI Horizons grant for partly supporting this research. Soumen Chakrabarti was partly supported by grants from IBM and SERB.

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

# CRUSH4SQL: Collective Retrieval Using Schema Hallucination For Text2SQL
# (Appendix)

## A   Anecdotes

In Table 14 we show examples of schema retrieved by baseline single embedding method and CRUSH. Observe how retrieved set from single embedding is biased towards matching one of the columns of the hallucinated schema.

## B   Prompts for the SocialDB dataset

In Table 15 we show the six few-shot examples in prompts for schema hallucination. The bottom half of the table shows four hallucinated schema obtained from the LLM.

| $x$ | What are the ids of students who both have friends and are liked? |
|---|---|
| $R(q(x))$ | network.friend.student_id, network.likes.liked_id |
| $R(x)$ | Single Embedding (OpenAI) |
| | college.student.id, 
 student_assessment.students.student_id, 
 school_player.school.school_id, 
 school_player.school_performance.school_id, 
 student_assessment.student_course_attendance.student_id, 
 student_assessment.candidate_assessments.candidate_id, 
 student_assessment.candidates.candidate_id, 
 voter.student.stuid 
 network_1.likes.student_id, 
 school_player.school.boys_or_girls |
| $R(x)$ | CRUSH |
| | student_assessment.students.student_id 
 network_1.likes.student_id 
 e_learning.students.student_id 
 network_1.friend.friend_id 
 network_1.friend.student_id 
 network_1.likes.liked_id 
 student_assessment.students.student_details 
 student_assessment.student_course_attendance.student_id 
 student_assessment.student_course_registrations.student_id 
 network_2.personfriend.friend] |

Table 14: Results from single embedding retrieval with OpenAI Vs CRUSH.

| # | LLM prompt: Hallucinate a minimal schema of a relational database that can be used to answer the natural language question. Here are some examples: |
|---|---|
| $x$ | What is the correlation between child nourishment and parental education in the state of Madhya Pradesh? |
| $\mathcal{K}$ | Family_health_survey(child age, child nourishment), Population_census( state, age-group, male literate population, female literate population) |
| $x$ | Health center per population ratio at the village level or district level from the year 2015? |
| $\mathcal{K}$ | Health_infrastructure(village, health care facility), Population_census(district, male population, female population) |
| $x$ | Distribution of medical professionals by type across regions from 2011 onwards from the state of Kerala. |
| $\mathcal{K}$ | Health_statistics_statewise(medical professional) |
| $x$ | Correlation between road connectivity and Mother Mortality Rate (MMR) during 2011 from the state UK. |
| $\mathcal{K}$ | Family_health_survey(state, year, maternal mortality), Road_statistics(state, road type) |
| $x$ | What is the trend for CPI of goods excluding food and fuel? |
| $\mathcal{K}$ | Inflation_money_and_credit(year, Categories of Consumer Expenditure) |
| $x$ | Correlation between number of bank branches and district growth? |
| $\mathcal{K}$ | Town_amenities_census(amenities, public works department), bank_details(number of branches, bank type) |
| **Hallucinated $\mathcal{K}$ generated by LLM given input $x$** | |
| $x$ | Which Central Public Sector Enterprise generated most employment in the 10 years? |
| $\mathcal{K}$ | employment_statistics(enterprise, year, employment), enterprise_details(enterprise, sector) |
| $x$ | how awareness among women impact births in caesarean section? |
| $\mathcal{K}$ | health_statistics_statewise(state, year, caesarean section births), women_awareness_survey(state, year, awareness level) |
| $x$ | what is the correlation between socio-economic status and health insurance enrollments? |
| $\mathcal{K}$ | socio_economic_status(income, education level), health_insurance_enrollment(age, gender, income level) |
| $x$ | Trend of Total Export Volume of Select Commodities to Principal Countries based on New commodity classification as per 2009-10 over a period of 5 years? |
| $\mathcal{K}$ | export_data(year, commodity, country, total export volume), commodity_classification(commodity, new commodity classification) |

Table 15: Examples of in-context training examples (for SocialDB) given to the LLM to prompt it to hallucinate a minimal schema of a database that can be used to answer the given question.

| # | LLM prompt: Hallucinate a minimal schema of a relational database that can be used to answer the natural language question. Here are some examples: |
|---|---|
| $x$ | What is the brand of the truck that is used to ship by Zachery Hicks? |
| $\mathcal{K}$ | truck(truck_id, make), shipment(truck_id, driver_id), driver(driver_id, first_name, last_name) |
| $x$ | State the name of the city where Jose Rodriguez works. |
| $\mathcal{K}$ | employee(locationID, firstname, lastname), location(locationID, locationcity) |
| $x$ | Please list all horror films that have a rating of 1. |
| $\mathcal{K}$ | u2base(movieid, rating), movies2directors(movieid, genre) |
| $x$ | List all the names of the books written by Danielle Steel. |
| $\mathcal{K}$ | book(book_id, title), book_author(book_id, author_id), author(author_id, author_name) |
| $x$ | How many female representatives are there in Michigan? |
| $\mathcal{K}$ | current(bioguide_id, bioguide, gender_bio), current_terms(bioguide, type, state) |
| $x$ | How many stars does each of the 3 top users with the most likes in their reviews have? |
| $\mathcal{K}$ | Tips(user_id, likes), Users(user_id, user_average_stars) |
| **Hallucinated $\mathcal{K}$ generated by LLM given input $x$** | |
| $x$ | Which country had the gas station that sold the most expensive product id No.2 for one unit? |
| $\mathcal{K}$ | Product(product_id, price), Gas_Station(gas_station_id, country), Sales(gas_station_id, product_id, quantity) |
| $x$ | Please list the titles of the posts owned by the user csgillespie? |
| $\mathcal{K}$ | Posts(post_id, title, user_id), Users(user_id, username) |
| $x$ | Which country is the constructor which got 1 point in the race No. 24 from? |
| $\mathcal{K}$ | race(race_id, constructor_id), points(race_id, constructor_id, points), constructor(constructor_id, country) |
| $x$ | What is the administrator's email address for the school with the highest number of test takers who received SAT scores of at least 1500?Provide the name of the school. |
| $\mathcal{K}$ | Schools(school_id, school_name, administrator_email), Test_takers(school_id, SAT_score) |

Table 16: Examples of in-context training examples (for BirdUnion) given to the LLM to prompt it to hallucinate a minimal schema of a database that can be used to answer the given question.