# OpenReview forum: "CRUSH4SQL: Collective Retrieval Using Schema Hallucination For Text2SQL"
_EMNLP/2023/Conference — EMNLP 2023 Main_

### Official Review · Reviewer_H9wu · 2023-07-25

**Soundness:** 4

**Excitement:**

4: Strong: This paper deepens the understanding of some phenomenon or lowers the barriers to an existing research direction.

**Paper Topic And Main Contributions:**

This paper addresses the problem of retrieving relevant schema elements from a large database schema to improve text-to-SQL systems. The main contributions are:

- Proposes a two-stage approach called CRUSH that first uses an LLM to hallucinate a minimal schema, then retrieves actual schema elements that collectively cover the hallucinated schema.
- Introduces two new large-schema benchmarks for evaluating retrieval-augmented text-to-SQL: SpiderUnion (4502 columns) and SocialDB (17.8k columns).
- Shows improved recall over baselines on these benchmarks, translating to gains in downstream SQL accuracy.

The core idea of harnessing LLM hallucination to bridge the gap between query phrases and schema elements is novel and promising. The collective optimization for high-recall retrieval is also well-motivated. The new benchmarks fill an important gap in text-to-SQL research.

**Questions For The Authors:**

- Could you provide any examples of how the hallucinated schemas differ from the gold schemas? How does this affect downstream SQL accuracy?
- Have you experimented with any techniques to make the hallucination prompts more robust and reduce reliance on gold schemas?
- Is there any plan to release the SocialDB benchmark to advance research in this area?
- Could you discuss any challenges faced in real-world deployment or limitations compared to lab setting?

**Reasons To Accept:**

- Novel technique for schema retrieval using LLM hallucination, which helps overcome lexical gaps between queries and schema.
- Principled objective function for collective retrieval that balances coverage and connectivity.
- Thorough experimental analysis demonstrating gains over baselines.
- New large-schema benchmarks that enable more realistic evaluation.
- Well-written paper that clearly explains the limitations of prior work, motivation for technical approach, and ablation studies.

**Reasons To Reject:**

- The techniques are evaluated only on the authors' new benchmarks. Additional experiments on other datasets could strengthen the results.
- The hallucination prompts are hand-designed based on gold schemas, which may limit real-world applicability. More analysis on robustness would help.

**Reproducibility:**

4: Could mostly reproduce the results, but there may be some variation because of sample variance or minor variations in their interpretation of the protocol or method.

**Reviewer Confidence:**

4: Quite sure. I tried to check the important points carefully. It's unlikely, though conceivable, that I missed something that should affect my ratings.

---

> ### Author Rebuttal · Authors · 2023-08-29
>
> Q1. *The techniques are evaluated only on the authors' new benchmarks. Additional experiments on other datasets could strengthen the results.*\
> Unavailability of large scale datasets for the public is our  reason for introducing new benchmarks for further research in this study.
> -> We are trying to run a second experiment by taking a union of the schema on another recently released Text-to-SQL benchmark.  We hope to publish the results soon.
>
> Q2. *The hallucination prompts are hand-designed based on gold schemas, which may limit real-world applicability. More analysis on robustness would help.*\
> Please note that the gold schema in our case is *very large*.  The few-shot examples cover only a small part of the gold schema.  Further, for the SpiderUnion datasets, the few-shot prompts were generated from the schema present in the training dataset whereas they were evaluated on unseen schema from the test set. We report additional numbers with other randomly chosen few-shot examples for the SpiderUnion dataset.  The results are present in the response to reviewer yafE.  We observe that our results are robust to many different ways of choosing the few-shots.
>
> Even with zero-shot prompting we get good schema hallucinations as seen in the additional experiments reported in the response to reviewer yafE
>
> Q3. *Could you provide any examples of how the hallucinated schemas differ from the gold schemas? How does this affect downstream SQL accuracy?*\
> We give three examples, two of which give rise to correct SQL and another example where hallucinated schema fails leading to gold schema not getting retrieved and thus wrong SQL
>
> Example 1:
> ```
> Question: Who owns the youngest dog? Give me his or her last name.
>
> Gold SQL: SELECT T1.last_name FROM Owners AS T1 JOIN Dogs AS T2 ON T1.owner_id  =  T2.owner_id WHERE T2.age  =  ( SELECT max(age) FROM Dogs )
>
> Gold Database schema:
> 	Breeds(breed_code, breed_name)
> 	Charges(charge_id, charge_type, charge_amount)
> 	Sizes(size_code, size_description)
> 	Treatment_Types(treatment_type_code, treatment_type_description)
> 	Owners(owner_id, first_name, last_name, street, city, state, zip_code, email_address, home_phone, cell_number)
> 	Dogs(dog_id, owner_id, abandoned_yn, breed_code, size_code, name, age, date_of_birth, gender, weight, date_arrived, date_adopted, date_departed)
> 	Professionals(professional_id, role_code, first_name, street, city, state, zip_code, last_name, email_address, home_phone, cell_number)
> 	Treatments(treatment_id, dog_id, professional_id, treatment_type_code, date_of_treatment, cost_of_treatment)
>
> Schema elements needed from Gold DB to write SQL:
> 	dogs(age, owner_id)
> 	owners(last_name, owner_id)
>
> Hallucinated Schema:
> 	owner(first name, last name, owner id)
> 	dog(dog id, age, owner id)
>
> CRUSH retrieval (top 10):
>        dog_kennels.dogs.owner_id
>        dog_kennels.dogs.dog_id
>        dog_kennels.owners.owner_id
>        dog_kennels.owners.first_name
>        dog_kennels.dogs.age
>        dog_kennels.owners.last_name
>        dog_kennels.dogs.name
>        dog_kennels.dogs.date_of_birth
>        dog_kennels.owners.state
>
> SQL (on CRUSH retrieval using chatGPT):
> SELECT o.last_name
> FROM owners o
> JOIN (
>     SELECT owner_id
>     FROM dogs
>     WHERE age = (SELECT MIN(age) FROM dogs)
>     LIMIT 1
> ) youngest_dog_owner
> ON o.owner_id = youngest_dog_owner.owner_id;
>
> Note: The Hallucinated Schema aligns closely with the "Schema Elements Needed to Write SQL." This facilitates more effective retrieval of relevant schema elements for constructing accurate SQL query.
>
> ```
>
> Example 2:
> ```
> Question: Give me the description of the treatment type whose total cost is the lowest.
>
> Gold SQL: SELECT T1.treatment_type_description FROM Treatment_types AS T1 JOIN Treatments AS T2 ON T1.treatment_type_code  =  T2.treatment_type_code GROUP BY T1.treatment_type_code ORDER BY sum(cost_of_treatment) ASC LIMIT 1
>
> Gold Database schema:
> 	Breeds(breed_code, breed_name)
> 	Charges(charge_id, charge_type, charge_amount)
> 	Sizes(size_code, size_description)
> 	Treatment_Types(treatment_type_code, treatment_type_description)
> 	Owners(owner_id, first_name, last_name, street, city, state, zip_code, email_address, home_phone, cell_number)
> 	Dogs(dog_id, owner_id, abandoned_yn, breed_code, size_code, name, age, date_of_birth, gender, weight, date_arrived, date_adopted, date_departed)
> 	Professionals(professional_id, role_code, first_name, street, city, state, zip_code, last_name, email_address, home_phone, cell_number)
> 	Treatments(treatment_id, dog_id, professional_id, treatment_type_code, date_of_treatment, cost_of_treatment)
>
> Schema elements needed from Gold DB to write SQL:
> 	treatment_types(treatment_type_code, treatment_type_description)
> 	treatments(cost_of_treatment, treatment_type_code)
>
> Hallucinated Schema:
> 	treatment_type(description, total cost)
>
> CRUSH retrieval (top 10):
>        dog_kennels.treatments.cost_of_treatment
>        hospital_1.trained_in.treatment
>        hospital_1.procedures.cost
>        hospital_1.medication.description
>        dog_kennels.treatment_types.treatment_type_description
>        dog_kennels.treatment_types.treatment_type_code
>        dog_kennels.treatments.treatment_type_code
>        dog_kennels.treatments.treatment_id
>        dog_kennels.treatments.date_of_treatment
>        dog_kennels.treatments.professional_id
>
> SQL (on CRUSH retrieval using chatGPT):
> SELECT tt.treatment_type_description
> FROM treatment_types tt
> JOIN treatments t ON tt.treatment_type_code = t.treatment_type_code
> GROUP BY tt.treatment_type_description
> HAVING SUM(t.cost_of_treatment) = (
>     SELECT MIN(total_cost)
>     FROM (
>         SELECT SUM(cost_of_treatment) AS total_cost
>         FROM treatments
>         GROUP BY treatment_type_code
>     ) AS subquery
> );
>
>
> Note: In this example, LLM hallucinates a minimal schema which aligns with the actual schema elements needed to write SQL and thus, leads to better retrieval.
> ```
>
> Example 3:
>
> ```
> Question: How many distinct nationalities are there?
>
> Gold SQL: SELECT count(DISTINCT Nationality) FROM people
>
> Gold Database schema:
> 	poker_player(Poker_Player_ID, People_ID, Final_Table_Made, Best_Finish, Money_Rank, Earnings)
> 	people(People_ID, Nationality, Name, Birth_Date, Height)
>
> Schema elements needed from Gold DB to write SQL:
> 	people(nationality)
>
> Hallucinated Schema:
> 	students(name, nationality)
>
> CRUSH retrieval (top 10):
>        student_assessment.students.student_details
>        student_assessment.students.student_id
>        e_learning.students.student_id
>        e_learning.students.personal_name
>        e_learning.students.middle_name
>        e_learning.students.family_name
>        e_learning.students.login_name
>        student_transcripts_tracking.students.other_student_details
>        student_transcripts_tracking.students.student_id
>        student_transcripts_tracking.students.ssn
>
> SQL (on CRUSH retrieval using chatGPT):
> SELECT COUNT(DISTINCT nationality) AS distinct_nationalities
> FROM students_student_assesment;
>
> Note: Given that the question becomes vague without dbid, and so does the hallucinated schema. Based on the few-shot examples LLM generates the best possible schema, and it would be wrong to expect LLM to incorporate any information regarding 'poker_player' in the hallucinated schema based solely on the question. Finally, with this hallucinated schema, it is difficult to get 'gold schema elements needed to write SQL' in topk, which has a direct consequence on the generated SQL
>
> ```
>
>
> Q4. *Have you experimented with any techniques to make the hallucination prompts more robust and reduce reliance on gold schemas?*\
> ->
> We do not depend much on gold schema to design the hallucination prompts.  For example, on the Spider datasets we found that even with zero-shot prompting we  hallucinated schema that is useful in retrieving the gold schema.
>
>
> Q5. *Is there any plan to release the SocialDB benchmark to advance research in this area?*\
> -> We plan to release the SocialDB benchmark after the anonymity period is over.
>
> Q6. *Could you discuss any challenges faced in real-world deployment or limitations compared to lab setting?*\
> -> Unlike academic benchmarks such as SPIDER and WikiSQL, real-world text-to-SQL datasets had a significantly higher number of columns per table, column names tended to be notably longer, and there was significant overlap in the words across column names.
> Furthermore, natural language utterances or questions posed within these datasets often fail to explicitly reference the schema items. This omission occurs because the database schema is not necessarily known to the user making the inquiry. For instance, the Spider dataset's question, "titles of films that include 'Deleted Scenes' in their special feature section," might be more naturally expressed as "films with deleted scenes" in a real-world scenario.
> Additionally, while datasets like SPIDER are extensive and span various domains, the queries within them are generated through crowdwork. Although these queries are based on real databases, they are essentially synthetic. This introduces a notable distinction from real-world datasets, where answering questions requires a substantial amount of real-world knowledge.

---

### Official Review · Reviewer_1yLK · 2023-08-04

**Soundness:** 4

**Excitement:**

4: Strong: This paper deepens the understanding of some phenomenon or lowers the barriers to an existing research direction.

**Missing References:**

No missing references in my opinion.

**Paper Topic And Main Contributions:**

The paper is on the topic of natural language question answering over SQL databases. The author(s) proposes a method called CRUSH that uses LLMs and dense retrieval to obtain a small subset of a database schema that can then be used as one of the inputs of a Text-to-SQL system. The first component of CRUSH uses the in-context learning functionality of GPT-3 to retrieve a (hallucinated) schema K(q) that could be used to answer a user question q. Then, K(q) is passed as input to an optimization problem which also takes as input elements of the real database schema that match K(q). The solution of the optimization problem is a small schema R(q) that can be used to answer q with an existing question answering engine. Another contribution of the paper is the creation of two benchmark datasets (SpiderUnion and SocialDB) that can be used to evaluate similar proposals. Finally, the paper contains an ablation study evaluating the effect of various components of CRUSH.

**Questions For The Authors:**

A. Lines 044-048. Can you give a reference that might be helpful to others for this claim?

B. Lines 228-231. Are you talking about your own unsuccessful attempts here? How about similar work by other researchers? Any papers with techniques that would not work here?

C. How did you select the examples that you use in your prompt to GPT-3 (Table 1)? What is special about them?



**Reasons To Accept:**

* The problem of natural language question answering over relational databases is an important one.
* The CRUSH framework presented by the paper is interesting and solves an important subproblem of the general question answering problem.
* The two benchmark datasets created by the author(s) are interesting in their own right.

**Reasons To Reject:**

No reasons to reject in my opinion.

**Reproducibility:**

5: Could easily reproduce the results.

**Reviewer Confidence:**

4: Quite sure. I tried to check the important points carefully. It's unlikely, though conceivable, that I missed something that should affect my ratings.

**Typos Grammar Style And Presentation Improvements:**

The paper is well written and it reads very nicely.
Some minor comments:
* Line 191: service
* Related work section. It is unusual to have the related work section before the experiments. In my experience, the related work section is either the second section or the section before the last in a paper.
* Line 384. RoBERTa

---

> ### Author Rebuttal · Authors · 2023-08-29
>
> Q1. *Can you give a reference that might be helpful to others for this claim?*
> 1. World Bank data (https://datacatalog.worldbank.org/home)
> 2. Data Commons (https://www.datacommons.org/)
> 3. One more reference is omitted due to the double-blind review process
>
> Q2. *Are you talking about your own unsuccessful attempts here? How about similar work by other researchers? Any papers with techniques that would not work here?*\
> -> Yes. We are not aware of any prior work on this scale of datasets for schema subsetting.
>
> Q3. *How did you select the examples that you use in your prompt to GPT-3 (Table 1)? What is special about them?*\
> -> We randomly sampled over the training dataset.  In the response to reviewer yafE we present results with multiple random selections.

---

### Official Review · Reviewer_yafE · 2023-08-04

**Soundness:** 4

**Excitement:**

4: Strong: This paper deepens the understanding of some phenomenon or lowers the barriers to an existing research direction.

**Paper Topic And Main Contributions:**

This paper addresses the problem of schema subsetting in large databases with tens of thousands of columns for Text-to-SQL generators. Encoding the entire schema with user text is expensive and impractical in such cases. The main challenge is that standard dense retrieval techniques are not suitable for ranking sets of schema elements rather than individual documents.

The paper makes the following contributions:

1. It proposes a novel two-stage process for schema subsetting in large databases, leveraging an LLM to hallucinate a minimal database schema.
2. It introduces two new benchmarks for schema subsetting on large databases.
3. The proposed method demonstrates significantly higher recall than state-of-the-art retrieval-based augmentation methods, indicating its potential for practical applications.

**Reasons To Accept:**

1. The paper presents a novel approach to schema subsetting in large databases, which can inspire further research and development in this area.

2. The introduction of new benchmarks provides valuable resources for evaluating and comparing future methods in schema subsetting.

3. The improved recall and enhanced Text-to-SQL accuracy demonstrated by the proposed method can contribute to the development of more efficient and accurate Text-to-SQL generators for large databases.

4. The paper's findings can potentially lead to more practical and cost-effective solutions for Text-to-SQL applications in real-world scenarios involving large databases with numerous columns.

**Reasons To Reject:**

- This paper relies a minimal database schema which is hallucinated by an LLM. It would be better to discuss the stability of this step, and how robust it is with respect to the final semantic parsing performance.

**Reproducibility:**

4: Could mostly reproduce the results, but there may be some variation because of sample variance or minor variations in their interpretation of the protocol or method.

**Reviewer Confidence:**

4: Quite sure. I tried to check the important points carefully. It's unlikely, though conceivable, that I missed something that should affect my ratings.

---

> ### Author Rebuttal · Authors · 2023-08-29
>
> Result 1: First we study the **effect of randomly shuffling the six few-shots**. We report mean and standard deviation of both recall for schema retrieval and accuracy of downstream text-to-sql generation.. Rest of the settings are the same as in the main tables in the paper.  Observe the low standard deviation in recall across the five runs.
> | Budget      | 3    | 5    | 10   | 20   | 30   | 50   |
> | ----------- | ---- | ---- | ---- | ---- | ---- | ---- |
> | SpiderUnion |      |      |      |      |      |      |
> | Recall Mean | 0.58 | 0.70 | 0.82 | 0.89 | 0.92 | 0.94 |
> | Recall Std  | 0.01 | 0.01 | 0.00 | 0.00 | 0.01 | 0.00 |
> | SocialDB    |      |      |      |      |      |      |
> | Recall Mean | 0.37 | 0.47 | 0.56 | 0.65 | 0.69 | 0.73 |
> | Recall Std  | 0.03 | 0.03 | 0.04 | 0.02 | 0.02 | 0.01 |
>
> EM/EX for different shuffles for SpiderUnion:
> | Budget  | 3    | 5    | 10   | 20   | 30   | 50   |
> | ------- | ---- | ---- | ---- | ---- | ---- | ---- |
> | EM Mean | 0.33 | 0.45 | 0.51 | 0.51 | 0.51 | 0.49 |
> | EM Std  | 0.02 | 0.00 | 0.01 | 0.01 | 0.02 | 0.02 |
> | EX Mean | 0.38 | 0.52 | 0.60 | 0.61 | 0.60 | 0.59 |
> | EX Std  | 0.01 | 0.00 | 0.01 | 0.02 | 0.02 | 0.02 |
>
> Result 2: Next we report **stability to different sets of examples chosen in the few-shots**.  Again observe the low standard-deviation in the resultant recall values.
> | Budget      | 3    | 5    | 10   | 20   | 30   | 50   |
> | ----------- | ---- | ---- | ---- | ---- | ---- | ---- |
> | SpiderUnion |      |      |      |      |      |      |
> | Recall Mean | 0.6  | 0.7  | 0.8  | 0.9  | 0.9  | 0.9  |
> | Recall Std  | 0.02 | 0.01 | 0.01 | 0.01 | 0.01 | 0.00 |
>
> EM/EX for different sets for SpiderUnion:
> | Budget  | 3    | 5    | 10   | 20   | 30   | 50   |
> | ------- | ---- | ---- | ---- | ---- | ---- | ---- |
> | EM Mean | 0.34 | 0.46 | 0.52 | 0.52 | 0.51 | 0.49 |
> | EM Std  | 0.01 | 0.02 | 0.01 | 0.02 | 0.02 | 0.02 |
> | EX Mean | 0.38 | 0.53 | 0.60 | 0.62 | 0.61 | 0.59 |
> | EX Std  | 0.01 | 0.01 | 0.01 | 0.02 | 0.02 | 0.02 |
>
> Result 3: Next we generate **multiple schemas from the LLM with different temperatures**.  The results do not change too much with temperature either.
> | Budget               | 3    | 5    | 10   | 20   | 30   | 50   |
> | -------------------- | ---- | ---- | ---- | ---- | ---- | ---- |
> | SpiderUnion          |      |      |      |      |      |      |
> | Recall at temp = 0   | 0.59 | 0.72 | 0.83 | 0.9  | 0.92 | 0.94 |
> | Recall at temp = 0.5 | 0.58 | 0.7  | 0.82 | 0.89 | 0.92 | 0.93 |
> | Recall at temp = 1   | 0.58 | 0.69 | 0.82 | 0.89 | 0.91 | 0.94 |
> | SocialDB             |      |      |      |      |      |      |
> | Recall at temp = 0   | 0.40 | 0.52 | 0.58 | 0.69 | 0.71 | 0.75 |
> | Recall at temp = 0.5 | 0.41 | 0.50 | 0.61 | 0.67 | 0.71 | 0.74 |
> | Recall at temp = 1   | 0.36 | 0.47 | 0.56 | 0.63 | 0.70 | 0.76 |
>
> Result 4: Finally, we observe that even with **zero-shots** the model provides significant gains over the baseline single-embedding based retrieval.
> | Budget              | 3    | 5    | 10   | 20   | 30   | 50   |
> | ------------------- | ---- | ---- | ---- | ---- | ---- | ---- |
> | SpiderUnion         |      |      |      |      |      |      |
> | zero shot           | 0.58 | 0.72 | 0.84 | 0.9  | 0.92 | 0.94 |
> | Single DPR (OpenAI) | 0.55 | 0.64 | 0.77 | 0.86 | 0.90 | 0.93 |
> | CRUSH (ours)        | 0.59 | 0.72 | 0.83 | 0.9  | 0.92 | 0.94 |
> | SocialDB            |      |      |      |      |      |      |
> | zero shot           | 0.33 | 0.43 | 0.52 | 0.65 | 0.68 | 0.74 |
> | Single DPR (OpenAI) | 0.39 | 0.44 | 0.49 | 0.56 | 0.60 | 0.67 |
> | CRUSH (ours)        | 0.40 | 0.52 | 0.58 | 0.67 | 0.69 | 0.71 |

---

### Meta-Review · Area_Chair_FBmJ · 2023-09-19

**Recommendation:** 5

**Metareview:**

The authors propose a novel usage of LLMs towards efficient schema subsetting in large databases. It also introduces large benchmarks for this purpose which can be used in future work. The reviewers concur on the utility of this work. It is also noted that additional experiments for their proposed technique with benchmarks not developed by the authors themselves would enrich the paper. However, this is a minor recommended update to the existing work. I would also recommend light restucturing of the work to highlight the most important results and numbers, as that gets a little lost in this existing presentation.

---

### Decision · Program_Chairs · 2023-10-07

**Decision:**

Accept-Main

**Comment:**

The authors propose a novel usage of LLMs towards efficient schema subsetting in large databases. It also introduces large benchmarks for this purpose which can be used in future work. The reviewers concur on the utility of this work. It is also noted that additional experiments for their proposed technique with benchmarks not developed by the authors themselves would enrich the paper. However, this is a minor recommended update to the existing work. I would also recommend light restucturing of the work to highlight the most important results and numbers, as that gets a little lost in this existing presentation.